# Screening for Hepatitis C Virus Reinfection Using a Behaviour-Based Risk Score among Men Who Have Sex with Men with HIV: Results from a Case–Control Diagnostic Validation Study

**DOI:** 10.3390/pathogens12101248

**Published:** 2023-10-16

**Authors:** Kris Hage, Marita van de Kerkhof, Anders Boyd, Joanne M. Carson, Astrid M. Newsum, Amy Matser, Marc van der Valk, Kees Brinkman, Joop E. Arends, Fanny N. Lauw, Bart J. A. Rijnders, Arne van Eeden, Marianne Martinello, Gail V. Matthews, Janke Schinkel, Maria Prins

**Affiliations:** 1Department of Infectious Diseases, Public Health Service of Amsterdam, 1018 WT Amsterdam, The Netherlands; 2Infectious Diseases, Amsterdam UMC Location, University of Amsterdam, 1105 AZ Amsterdam, The Netherlands; 3Infectious Diseases, Amsterdam Institute for Infection and Immunity, 1105 AZ Amsterdam, The Netherlands; 4Stichting SBOH, 3528 BB Utrecht, The Netherlands; 5Stichting HIV Monitoring (SHM), 1105 BD Amsterdam, The Netherlands; 6The Kirby Institute, University of New South Wales, Sydney 2052, Australia; 7Department of Internal Medicine, Onze Lieve Vrouwe Gasthuis (OLVG), 1091 AC Amsterdam, The Netherlands; 8Department of Internal Medicine and Infectious Diseases, University Medical Center Utrecht (UMCU), 3584 CX Utrecht, The Netherlands; 9Department of Health, Medicine and Life Sciences, Maastricht University, 6211 LK Maastricht, The Netherlands; 10Department of Internal Medicine, Medical Centre Jan van Goyen, 1075 HN Amsterdam, The Netherlands; 11Department of Internal Medicine and Infectious Diseases, Erasmus University Medical Center, 3015 GD Rotterdam, The Netherlands; 12Department of Internal Medicine, DC Klinieken Oud Zuid, 1075 BG Amsterdam, The Netherlands; 13Medical Microbiology and Infection Prevention, Amsterdam UMC Location, University of Amsterdam, 1105 AZ Amsterdam, The Netherlands

**Keywords:** HCV, HCV reinfection, risk behaviour, MSM, HIV/hepatitis C virus coinfection

## Abstract

We assessed the predictive capacity of the HCV-MOSAIC risk score, originally developed for primary early HCV infection, as a screening tool for HCV reinfection in 103 men who have sex with men (MSM) with HIV using data from the MOSAIC cohort, including MSM with HIV/HCV-coinfection who became reinfected (cases, *n* = 27) or not (controls, *n* = 76) during follow-up. The overall predictive capacity of the score was assessed using the area under the receiver operating characteristic (AUROC) curve. The effects of covariates on the receiver operating characteristic (ROC) curve were assessed using parametric ROC regression. The score cut-off validated for primary early infection (≥2.0) was used, from which the sensitivity and specificity were calculated. The AUROC was 0.74 (95% confidence interval (CI) = 0.63–0.84). Group sex significantly increased the predictive capacity. Using the validated cut-off, sensitivity was 70.4% (95%CI = 49.8–86.2%) and specificity was 59.2% (95%CI: 47.3–70.4%). External validation from a cohort of 25 cases and 111 controls, all MSM with HIV, resulted in a sensitivity of 44.0% (95%CI = 24.4–65.1) and specificity of 71.2% (95%CI = 61.8–79.4). The HCV-MOSAIC risk score may be useful for identifying individuals at risk of HCV reinfection. In sexual health or HIV-care settings, this score could help guide HCV-RNA testing in MSM with a prior HCV infection.

## 1. Introduction

Since the 2000s, men who have sex with men (MSM) with HIV have witnessed a large increase in hepatitis C virus (HCV) infections [1,2]. With the introduction of direct-acting antivirals in 2014, treatment uptake rapidly increased. Consequently, strong decreases in HCV incidence have been observed, particularly in MSM with HIV [3,4]. Nevertheless, individuals who continue to engage in activities associated with HCV acquisition can become reinfected after successful treatment or spontaneous clearance (SC). In fact, over the past five years, more than half of HCV infections in MSM with HIV living in Western Europe have been reinfections [4,5]. Finding those with HCV reinfection as early as possible is therefore needed, which could reduce onward transmission. 

After primary HCV infection, anti-HCV antibodies develop and continue to circulate in the blood for years, even after treatment-induced sustained virologic response (SVR) or SC. Therefore, HCV-reinfected individuals can only be identified by a positive HCV-RNA test result. However, HCV-RNA testing is costly. Given the low numbers of new HCV cases among MSM in the Netherlands, routinely testing all MSM with HIV might not be the most cost-effective strategy to identify HCV infections, and case-finding based on reported behavioural risk of HCV infection has become more important [6,7]. Targeting HCV-RNA testing to only those individuals at high risk for HCV reinfection could help reduce the costs associated with excessive testing and identify those in need of immediate screening. 

Previously, Newsum et al. developed and validated a behaviour-based risk score for primary early HCV infection to help guide HCV testing in MSM with HIV [8]. The HCV-MOSAIC risk score consists of six factors associated with HCV acquisition, i.e., condomless receptive anal intercourse (RAI), sharing sex toys, unprotected fisting, injecting drug use (IDU), sharing snorting paraphernalia, and having an ulcerative sexually transmitted infection (STI). Primary early HCV infection could be identified with high sensitivity (Se) and specificity (Sp) in MSM with HIV and an AUROC of 0.82 [8]. In the current study, we aimed to assess the predictive capacity of the HCV-MOSAIC risk score as a screening tool for HCV reinfection in MSM with HIV specifically (using data from the same cohort) and to externally validate the score in a setting with similar epidemiology of HCV infection.

## 2. Materials and Methods

### 2.1. Study Design

We conducted a case–control diagnostic validation study of the HCV-MOSAIC score for the outcome of HCV reinfection using data from two prospective cohort studies. The training dataset was obtained from participants enrolled in the prospective Dutch MSM Observational Study of Acute Infection with hepatitis C (MOSAIC) between 2009 and 2017. MSM with HIV and an acute HCV infection were enrolled, provided that they had had a confirmed acute HCV infection prior to inclusion. Sociodemographic, clinical, and virological data for HIV and HCV were retrospectively collected from primary HCV infection and prospectively collected at each semi-annual visit following inclusion, together with an extensive self-administered questionnaire containing questions about risk behaviour referring to the preceding 6 or 12 months. The MOSAIC study was approved by the Institutional Review Boards of the Amsterdam UMC (University of Amsterdam, Amsterdam, the Netherlands) and boards of directors at all six participating centres. All participants gave written informed consent, and the study was conducted according to hospital ethical guidelines and the 2011 Dutch code of conduct for responsible use of human tissue and medical research [9].

After extensive search for an external validation dataset, only one study among MSM at risk of reinfection with sufficient behavioural data was identified. The Recently Acquired HCV Infection Trial (REACT) was an international open-label, phase III, randomised trial among individuals with a recent HCV infection between 2017 and 2019 (https://clinicaltrials.gov/study/NCT02625909?term=NCT02625909&rank=1 (accessed on 10 July 2023), identifier: NCT02625909) [10]. The external validation dataset was obtained from all individuals enrolled in this cohort. Participants completed questionnaires every three months for up to two years, including sociodemographic, clinical, and virological data and data on risk behaviours referring to the preceding month.

### 2.2. Participant Selection

For the present study, participants were considered from the visit at which they achieved SC, defined as having two negative HCV-RNA tests following a positive HCV-RNA test in untreated patients, or SVR following primary HCV infection, defined as at least one negative HCV-RNA test 12 or 24 weeks after the end of treatment (depending on treatment regimen) or SVR as indicated in the patient’s medical file. We identified participants who became reinfected with HCV (cases), defined as having detectable HCV-RNA after achieving spontaneous clearance or SVR following treatment, and those who did not have HCV reinfection (controls), defined as at least two consecutive HCV-RNA negative visits while being at risk of HCV reinfection during a minimum follow-up of 6 months. In the analysis, we selected the visit closest to the estimated date of reinfection for cases and a randomly selected visit during follow-up for controls. Estimated date of reinfection was calculated using the midpoint assumption between the last negative and the first positive HCV-RNA test. The same definitions for cases and controls were used for both the training and external validation datasets.

### 2.3. Statistical Analysis

The HCV-MOSAIC risk score was calculated for all included participants (Appendix A Appendix A). Briefly, this score was derived from a multivariable logistic regression model using data from the Dutch MOSAIC study (2009–2013) [8]. The score is calculated by summing the beta coefficients of the following six factors, when present: condomless RAI (beta 1.1), sharing sex toys (beta 1.2), unprotected fisting (beta 0.9), IDU (beta 1.4), sharing straws during nasally administered drug use (beta 1.0), and ulcerative STI (beta 1.4). In the training dataset, data on these behaviours were obtained from self-administered questionnaires and questions about risk behaviour referred to the preceding 6 or 12 months. In the external validation dataset, data on these behaviours were also collected through self-administered questionnaires. However, data on two of the variables in the risk score (sharing sex toys and unprotected fisting) were not collected and therefore not scored. In sensitivity analyses, we restricted the HCV-MOSAIC risk score in the training dataset to the same risk factors measured in the validation dataset to ensure comparability. 

To assess the overall predictive capacity of the continuous HCV-MOSAIC score for HCV reinfection, the area under the receiver operating characteristic (AUROC) curve was estimated along with 95% confidence intervals (CIs). To assess whether certain determinants could influence the receiver operating characteristic curve (ROC), parametric ROC regression was used. The parametric ROC curve was modelled as a normal cumulative distribution function combining a linear predictor of determinants and an inverse normal function of 1-Sp [2]. From this model, the average difference in ROC curve function between levels of covariates can be estimated across 1-Sp (ΔROC). To estimate 1-Sp, we used an inverse normal function conditioned on the covariate age. The 95%CIs of ΔROC were obtained from variance estimations using 1000 bootstrapped replications with replacement. We included covariates in the model that were proximal to HCV transmission (i.e., unrelated to the variables in the HCV-MOSAIC risk score or HCV transmission in general). The determinants that were individually tested were the number of HCV reinfections (categorised as ≤1 reinfection versus multiple reinfections), any anonymous partner, and any group sex in the preceding 6 months. 

The optimal cut-off for the HCV-MOSAIC risk score for primary early HCV infection was ≥2.0 [8]. Using MOSAIC data, the Se and Sp of predicting HCV reinfection were calculated using this cut-off, along with the proportion needed to be tested. To determine whether the cut-off needed to be recalibrated for HCV reinfection, we conducted a post hoc analysis in which an optimal cut-off was chosen at or above the score yielding the highest (Se + Sp)/2 and the proportion of correctly classified individuals. Additionally, the proportion needed to be tested was calculated at this cut-off. 

Statistical analyses were conducted using Stata 15.0 (Stata Corp, College Station, TX, USA). *p* < 0.05 was considered statistically significant.

## 3. Results

### 3.1. Description of the Study Population

Of the 143 participants with primary HCV infection enrolled in the MOSAIC study, 103 MSM with HIV resolved their HCV infection and were included in the training dataset in the analysis. Of those, 27 were identified as cases and 76 as controls, with a median follow-up time of 2.5 years (IQR = 1.0–4.7). A description of the sociodemographic characteristics and HCV-MOSAIC risk score variables of the study participants is provided in Table 1.

In the external validation dataset, 136 MSM with HIV cleared their primary HCV infection and were included in the validation analyses. Of those, 25 were identified as cases and 111 as controls, with a median follow-up time of 1.3 years (IQR = 0.8–1.6) (Table 1).

### 3.2. Performance of the HCV-MOSAIC Score for Reinfection in the Training Dataset

The median HCV-MOSAIC risk score was 2.5 (IQR = 1.2–3.4) for cases and 1.1 (IQR = 0.0–2.3) for controls (*p* < 0.001). The AUROC for predicting HCV reinfection was estimated at 0.74 (95%CI = 0.63–0.84) (Figure 1 and Table 2). In the ROC regression analysis, group sex (ΔROC = 1.14, 95%CI: 0.19–2.09) had a significant effect on the ROC curve at any given 1-Sp (Figure 2). There was no significant effect on the ROC curve with the number of HCV reinfections (ΔROC = 0.02, 95%CI: −0.87, 0.90) or any anonymous partner (ΔROC = 0.42, 95%CI: −1.18, 2.01). 

Using the validated cut-off for primary HCV infection, Se was 70.4% (95%CI = 49.8–86.2), Sp was 59.2% (95%CI = 47.3–70.4), and the proportion correctly classified was at 0.62. The proportion to be tested (i.e., the proportion of cases and controls with a risk score ≥ 2.0) was 48.5% (Table 2). Of note, Se was 100.0%, Sp was 29.4%, and the proportion correctly classified was at 0.51 when including only those who engaged in group sex. In a post hoc analysis using complete data, an optimal cut-off ≥1.2 was observed for this study population, at which Se was 77.8% (95%CI = 57.7–91.4), Sp was 57.9% (95%CI = 46.0–69.1), and the proportion correctly classified was at 0.63. The proportion to be tested (i.e., the proportion of cases and controls with a risk score ≥ 1.2) was 51.5%. 

### 3.3. Performance of the HCV-MOSAIC Score for Reinfection in the External Validation Dataset

In the external validation dataset, the median HCV-MOSAIC risk score was 1.1 (IQR = 1.1–2.5) for cases and 1.1 (IQR = 0.2–2.1) for controls (p = 0.357). The AUROC for predicting HCV reinfection was estimated at 0.63 (95%CI = 0.53–0.74) (Figure 3). The Se and Sp of the HCV-MOSAIC risk score were 44.0% (95%CI = 24.4–65.1) and 71.2% (95%CI = 61.8–79.4), respectively, at the validated cut-off for primary HCV infection (Table 2). The proportion correctly classified was at 0.66. Using a cut-off ≥1.2, Se was 44.0% (95%CI = 24.4–65.1), Sp was 66.7% (95%CI = 57.1–75.3), and the proportion correctly classified was at 0.63.

Sensitivity analyses where we restricted the HCV-MOSAIC risk score in the training dataset to the same risk factors measured in the validation dataset yielded comparable results compared to the external validation study at either cut-off (Appendix A and Appendix B). 

## 4. Discussion

The HCV-MOSAIC risk score may be able to identify individuals at high risk for HCV-reinfection with a slightly lower AUROC than that of primary early HCV-infection, for which the score was originally developed [8]. Using a cut-off value of ≥2.0, Se was 70.4% and Sp was 59.2%. Furthermore, 48.5% of MSM with a history of HCV would be advised to undergo HCV-RNA testing. 

When the HCV-MOSAIC risk score was developed for primary early HCV infection, Se and Sp of 78.0% and 78.6%, respectively, were observed [8]. In our study, a cut-off ≥2.0 resulted in decreased Se and Sp (70.4% and 59.2%, respectively), potentially reducing the clinical usefulness of the instrument. Further calibration in this population led to a cut-off ≥ 1.2 and yielded similar Se and AUROC of this risk score to identify HCV reinfection among MSM with HIV compared to the developmental study [8]. However, Sp was lower regardless of the cut-off, indicating that more individuals who are not HCV-reinfected will be identified for further testing. As ongoing HCV transmission is still a concern and HCV reinfection incidence is high in the Netherlands in this subgroup of individuals, case-finding to identify as many HCV reinfections as possible would be ideal [3]. This approach would require higher Se for screening, which is the case for the HCV-MOSAIC score at either cut-off. 

Interestingly, group sex significantly affected the ROC curve, increasing its predictive capacity. Using this score for MSM engaging in group sex also greatly increased the Se for screening HCV reinfection. Group sex may be associated with risk behaviours included in the risk score, including condomless RAI, IDU, and sharing snorting paraphernalia [11,12]. Sexualised drug use, a practice somewhat common during group sex [11], may prolong sexual interactions and hence increase the risk of reinfection [13,14,15].

It should be stressed that external validation of this risk score is complicated by the dearth of data existing on these specific behaviours. After an extensive search, the only study that could be remotely used for external validation was the REACT study [10]. This validation comes with the caveat that not all data on the included variables were available, most likely resulting in an underestimated risk score for some participants, and any inference on validity should be viewed as approximate. We found a somewhat lower AUROC in this external validation dataset than in the MOSAIC study, which suggests a slightly decreased capacity to identify individuals with HCV reinfection. Furthermore, since the Se was substantially lower in REACT compared to the training dataset, a large proportion of individuals with HCV reinfection would not have been screened using the risk score despite actually having an infection. Cohort studies with high numbers of HCV reinfections collecting detailed behavioural data, such as those in the HCV-MOSAIC risk score, would be valuable to further validate and calibrate the HCV-MOSAIC risk score for HCV reinfection. 

For primary HCV infection, the European AIDS Clinical Society recommends HCV screening with an anti-HCV antibody test at the time of HIV diagnosis and annually thereafter. Those who report engaging in sexual activities associated with HCV transmission should be tested for HCV infection every 3 to 6 months. For HCV reinfection, it is stated that “HCV-RNA or HCV core-antigen testing is also recommended in persons with ongoing risk behaviour for HCV re-infection after successful treatment or spontaneous clearance at 3 to 6-monthly intervals” [16]. As clinicians might find it difficult to appropriately assess which risk behaviours merit further testing for either primary infection or reinfection [16], we have shown previously [8] and in this study that this score could be useful to guide them in determining the risk necessary for additional HCV testing, particularly among those with higher risk scores. This can help shorten the time taken to diagnose HCV reinfection, prevent treatment delays, and help reach HCV micro-elimination targets. For MSM with HIV with lower risk scores, testing less frequently than annually, possibly every two to three years, may be considered. Additionally, the risk score could be used as an instrument to identify those who would benefit most from behavioural interventions aimed at preventing HCV reinfection.

This study is not without limitations. First, the risk factors were based on the most predictive risk factors found within the MOSAIC study population. Other risk factors could also bear predictive usefulness in other populations, among which are a lower nadir CD4 cell count [17], ALT levels [18], and a higher number of casual sex partners [17,19,20]. However, their strength of association is inconsistent across studies, and some factors are not suitable for implementation as they are not routinely available (e.g., CD4 cell count and ALT measurements at STI clinics) [8,21]. Second, because we selected the visit closest to the estimated date of reinfection, reporting bias may have occurred if the selected visit was after HCV reinfection. Finally, sample sizes were relatively small, and thus the risk score validation for HCV reinfection might not be entirely robust. Despite these limitations, the HCV-MOSAIC risk score may be helpful to identify MSM with HIV who are at risk for HCV reinfection in addition to recommended screening practices. 

## Figures and Tables

**Figure 1 pathogens-12-01248-f001:**
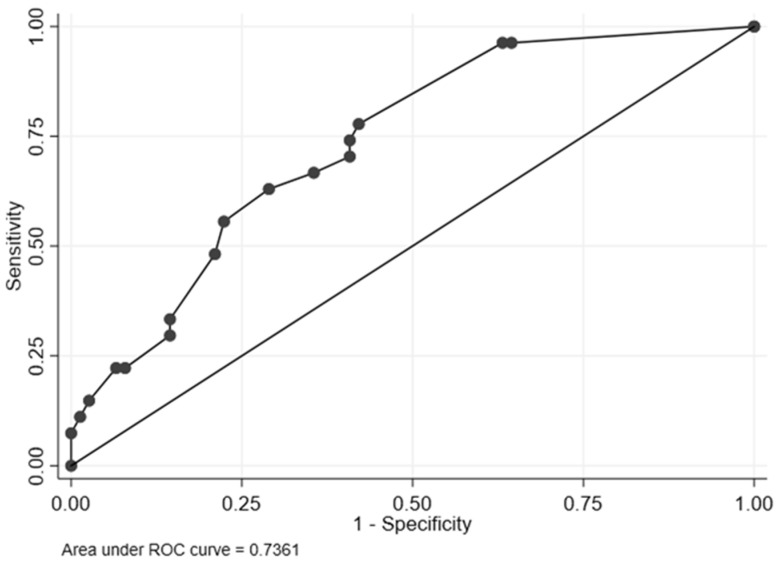
Non-parametric receiver operating characteristic (ROC) curve of the HCV-MOSAIC risk score for reinfection in the training dataset. ROC, receiver operating characteristic.

**Figure 2 pathogens-12-01248-f002:**
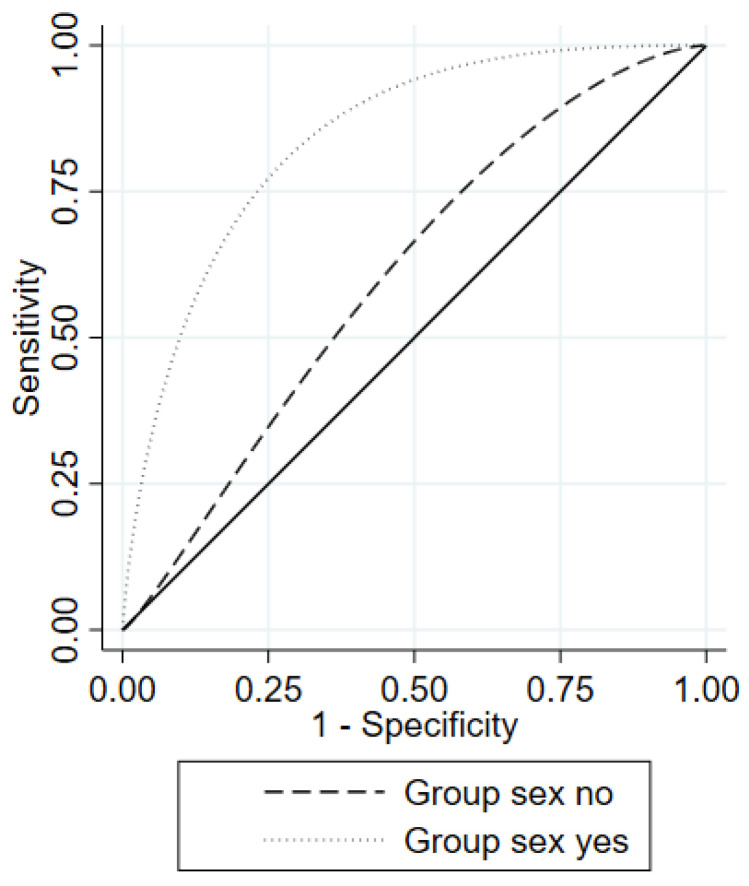
Parametric ROC curve for the HCV-MOSAIC risk score according to engaging in group sex in the training dataset.

**Figure 3 pathogens-12-01248-f003:**
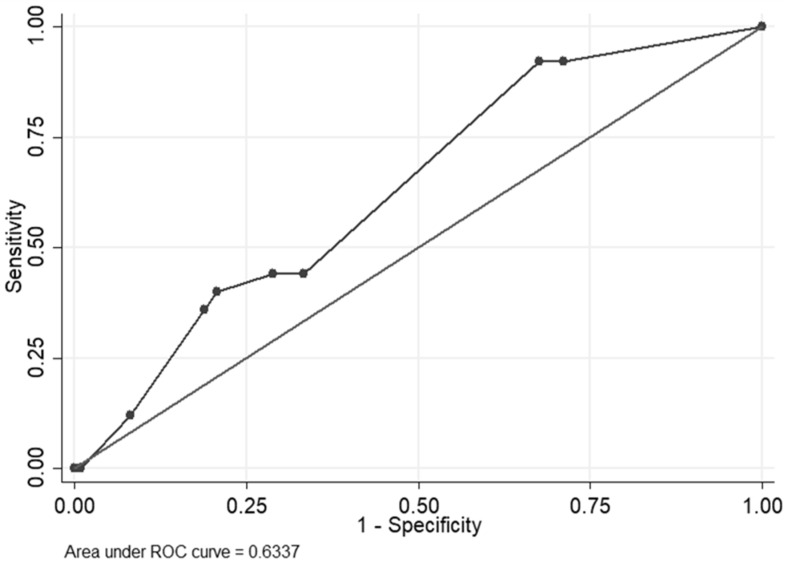
Non-parametric receiver operating characteristic (ROC) curve of the HCV-MOSAIC risk score for reinfection in the external validation dataset. ROC, receiver operating characteristic.

**Table 1 pathogens-12-01248-t001:** Characteristics of MSM with HIV with a previous HCV infection in the training and external validation studies.

	Training Dataset (MOSAIC Study, The Netherlands)			External Validation Dataset (REACT Study, Australia)		
	Cases: HCV Reinfection (n = 27)	Controls: No Reinfection (n = 76)	*p*	Cases: HCV Reinfection (n = 25)	Controls: No Reinfection (n = 111)	*p*
**Sociodemographic characteristics**						
Age, median (IQR)	42.4 (38.7–49.8)	47.9 (44.3–51.9)	0.035	47.3 (41.5–52.2)	44.9 (38.6–51.4)	0.659
Ethnicity, n (%) ^†^						
Dutch	23 (85.2)	60 (79.0)	0.700	5 (20.0)	16 (14.4)	0.541
Non-Dutch	4 (14.8)	15 (19.7)		20 (80.0)	95 (85.6)	
Educational level, n (%) ^‡^						
Low and middle	8 (29.6)	22 (29.0)	0.835	13 (52.0)	52 (46.8)	0.664
High	19 (70.4)	53 (69.7)		12 (48.0)	59 (53.2)	
**HCV-MOSAIC risk score** ^§^ [1]						
Condomless RAI, n (%) ^6M^	24 (88.9)	47 (61.8)	0.009	22 (88.0)	67 (60.4)	0.010
Sharing of sex toys, n (%) ^6M^	13 (48.2)	15 (19.7)	0.004	NA	NA	NA
Unprotected fisting, n (%) ^6M^	13 (48.2)	19 (25.0)	0.026	NA	NA	NA
Injecting drug use, n (%) ^12M^	5 (18.5)	3 (4.0)	0.015	9 (36.0)	23 (20.7)	0.120
Sharing of straws when NAD used, n (%) ^12M^	7 (25.9)	11 (14.5)	0.178	5 (20.0)	24 (21.6)	1.000
Ulcerative STI, n (%) ^12M^	4 (14.8)	6 (7.9)	0.297	1 (4.0)	6 (5.4)	1.000
Risk score, median (IQR)	2.5 (1.2–3.4)	1.1 (0–2.3)	<0.001	1.1 (1.1–2.5)	1.1 (0.2–2.1)	0.357

Abbreviations: HCV, hepatitis C virus; IQR, interquartile range; NA, not applicable; NAD, nasally administered drug; STI, sexually transmitted infection; RAI, receptive anal intercourse; REACT, Recently Acquired HCV Infection Trial; 6M, during the past 6 months; 12M, during the past 12 months; ^†^ in the MOSAIC study: for one control, information about ethnicity was missing; ^‡^ in the MOSAIC study: for one control, information about educational level was missing; ^§^ risk factors were measured at the visit closest to the estimated date of reinfection for cases and at the randomly selected visit during follow-up for controls.

**Table 2 pathogens-12-01248-t002:** Performance of the HCV-MOSAIC risk score for HCV reinfection in the training and external validation datasets using the validated cut-off ≥2.0.

	Training Dataset (MOSAIC Study, The Netherlands)	External Validation Dataset (REACT Study, Australia)
Sensitivity (95%CI)	70.4% (49.8–86.2)	44.0% (24.4–65.1)
Specificity (95%CI)	59.2% (47.3–70.4)	71.2% (61.8–79.4)
Proportion to be tested ^†^	48.5%	31.6%
AUROC (95%CI)	0.74 (0.63–0.84)	0.63 (0.53–0.74)

Abbreviations: AUROC, area under the receiver operating characteristic; CI, confidence interval; HCV, hepatitis C virus; MOSAIC; REACT, Recently Acquired HCV Infection Trial. ^†^ Proportion of all cases and controls with a risk score ≥ 2.0.

## Data Availability

Data are available upon reasonable request from the Principal Investigator.

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
