# Peer review of "Screening for Hepatitis C Virus Reinfection Using a Behaviour-Based Risk Score among Men Who Have Sex with Men with HIV: Results from a Case–Control Diagnostic Validation Study"

_pathogens, 2023, doi:10.3390/pathogens12101248_

Round 1
Reviewer 1 Report
Thank you for the opportunity to review this interesting manuscript. I express my congratulations to the authors for their initiative. It seems that having sexually transmitted and blood-borne diseases, having sex with multiple partners, and performing unprotected anal sex increased the risk of HCV infection in the transgender population. In the current study, authors assessed the predictive capacity of the HCV-MOSAIC risk score as a screening tool for specifically HCV reinfection in men who have sex with men with HIV.
I have only some suggestion. In the title, I would make it clear that it is a cross-sectional study. The statistical software used should be mentioned in the methods, which are well-organized e systematically. The discussion cold be implemented.
I think we should discuss how we use that score because HCV is one of the diseases that should be routinely checked in HIV patients. Should there be a difference in scope? Please, I request that the authors provide additional information about the risk score application.
Reviewer 2 Report
Dear Authors,
It is understood that in your study titled "Screening for hepatitis C virus reinfection using a behaviour-based risk score among men who have sex with men with HIV", you aim to evaluate the HCV risk assessment based on risk scoring. However, HCv is one of the diseases that should be routinely checked in HIV patients. These patients are in the disease group that should be checked in the presence of risky behaviors or routinely annually. Risk scoring is an application that can be made for non-routine examinations and evaluations. Even if a detailed statistical evaluation was made in your study, scoring for HCV screening and evaluation in HIV patients is not necessary, moreover, it is not accurate, it should be done routinely at certain times.
The fact that there was a patient and a control group from two different countries in your study also caused differences in the data of the study. As can be seen, different societies may have different behavioral models. However, HIV patients are in the risk group for HCV no matter which country they live in and should be routinely screened at certain periods. Social differences can make a difference in the incidence of HCV.
As a result, even though a scientific evaluation was made in the current study, I suggest rejecting the study because it is not correct in terms of practical application.
Best regards
Reviewer 3 Report
The authors used the HCV-MOSAIC risk score, originally for primary early HCV infection, as a screening tool for HCV re-infection. With the score cut-off of >=2, the sensitivity and specificity are 70.4% and 59.2%, respectively, with AIROC of 0.74; with the score cut-off of >= 1.2, the sensitivity and specificity become 77.8% and 57.9%, respectively, with AIROC of 0.63.
- Because the sensitivity of 77.8% with the cut-off of >=1.2 is higher than that of 70.4% with the cut-off of >= 2, authors may consider presenting the performance of HCV-MOSAIC risk score with he cut-off of >=1.2 in REACT study, Australia.
- With all the limitations of the risk score mentioned in the study, regular testing is the most effective way to identify patents with HCV-reinfection. The authors may elaborate more regarding what populations to use the risk score instead of testing.
- Given the high incidence of HCV reinfection in PLWH with HCV clearance by DAA or spontaneously, I wonder if the performance of the HCV-MOSAIC risk score would be different in this population.
Round 2
Reviewer 2 Report
Dear Authors,
I have revealed my view about your study as before. HCV should be screened routinely in HIV patients once a year without any risk stratification. Best regards
